# Existence and features of the myodural bridge in Gentoo penguins: A morphological study

**Cheng Chen**[1], **Sheng-bo Yu**[1], **Yan-yan Chi**[1], **Guang-yuan Tan**[2], **Bao-cheng Yan**[2], **Nan Zheng**[1]*, **Hong-Jin Sui**[1,3]*

**1** Department of Anatomy, College of Basic Medicine, Dalian Medical University, Dalian, China, **2** Haichang Ocean Park Holdings., Ltd, Biological Conservation Center, Shanghai, China, **3** Dalian Hoffen Preservation Institution, Dalian, China

* suihj@hotmail.com (HJS); zhengnan831016@163.com (NZ)

**Data Availability Statement:** All relevant data are within the manuscript and its Supporting information files.

**Funding:** This research was supported by the National Natural Science Foundation of China

## Abstract

Recent studies have evidenced that the anatomical structure now known as the myodural bridge (MDB) connects the suboccipital musculature to the cervical spinal dura mater (SDM). In humans, the MDB passes through both the posterior atlanto-occipital and the posterior atlanto-axial interspaces. The existence of the MDB in various mammals, including flying birds (*Rock pigeons and Gallus domesticus*) has been previously validated. Gentoo penguins are marine birds, able to make 450 dives per day, reaching depths of up to 660 feet. While foraging, this penguin is able to reach speeds of up to 22 miles per hour. Gentoo penguins are also the world's fastest diving birds. The present study was therefore carried out to investigate the existence and characteristics of the MDB in Gentoo penguin (*Pygoscelis papua*), a non-flying, marine bird that can dive. For this study, six Gentoo penguin specimens were dissected to observe the existence and composition of their MDB. Histological staining was also performed to analyze the anatomic relationships and characteristic of the MDB in the Gentoo penguin. In this study, it was found that the suboccipital musculature in the Gentoo penguin consists of the rectus capitis dorsalis minor (RCDmi) muscle and rectus capitis dorsalis major (RCDma) muscle. Dense connective tissue fibers were observed connecting these two suboccipital muscles to the spinal dura mater (SDM). This dense connective tissue bridge consists of primarily type I collagen fibers. Thus, this penguin's MDB appears to be analogous to the MDB previously observed in humans. The present study evidences that the MDB not only exists in penguins but it also has unique features that distinguishes it from that of flying birds. Thus, this study advances the understanding of the morphological characteristics of the MDB in flightless, marine birds.

## Introduction

The myodural bridge (MDB) is an anatomical structure connecting a suboccipital muscle (RCPmi) to the cervical spinal dura mater (SDM) in humans was identified in the atlanto-occipital interspace by Hack et al. (1995) [1]. Each myodural bridge (MDB) is now described as a fibrous, dense structure connecting the suboccipital musculature to the SDM, including

(NSFC31871213 to HJS); and Department of Education of Liaoning Province (LZ2020048 to NZ). The funders had no role in study design, data collection and analysis, decision to publish, or preparation of the manuscript.

**Competing interests:** Haichang Ocean Park Holding Co., Ltd. is responsible for the breeding of Gentoo Penguin (Pygoscelis Papua) and provides the basic data related to the penguin specimens for this research group. Another part of their work is the study and popularization of science related to penguins. The authors have declared that no competing interests exist. This does not alter our adherence to PLOS ONE policies on sharing data and materials.

fibers originating from the rectus capitis posterior minor (RCPmi) muscle, the rectus capitis posterior major (RCPma) muscle, the obliques capitis inferior (OCI) muscle, and the nuchae ligament (NL) [2–7, 13, 14]. It has been proposed that the MDB's fibers connecting the suboccipital muscles to the cervical SDM [1] might prevent dural infolding, thus maintaining the normal flow of the cerebrospinal fluid within the cisterna magna or cerebellomedullaris cistern [4, 7–14]. Sui et al. proposed that the MDB may play an important role in modulating the circulation of cerebrospinal fluid [9]. According to also reports, MDB dysfunction may occur with pathological conditions of the RCPmi muscle, resulting in the generation of cervicogenic headache, and other craniofacial disorders [15–19].

In recent years, researchers have found that the MDB is a universal structure in mammals [20], including marine mammals (*Nephocaena phocaenoides* and sperm whales) [21, 22]. The MDB also exists in reptiles (Siamese crocodile and *Trachemys scripta elegans*) [23, 24], as well as in birds (Rock pigeons and *Gallus domesticus*) [25, 26]. Therefore the universal existence of the MDB among reptiles, birds, and mammals suggests that the MDB is an evolutionarily conserveed anatomic structure having important biological functions. Moreover, we suggest that there may be structural differences in among various animals to adapt to different environmental conditions.

Penguins are unique birds that can both walk on land and dive and swim in deep water. Penguins are flightless birds, unlike the Rock pigeon and the *Gallus domesticus*. This marine bird's ability to swim presents it with different living environments and habits, then those of non-marine birds. Therefore, Gentoo penguins (*Pygoscelis papua*) will be studied in this paper. This study will provide a comparative anatomical foundation for the future study of the MDB in different animals.

## Materials and methods

For this scientific research study, six adult Gentoo penguin specimens from Haichang Ocean Park Holdings., Ltd., that died of natural causes, were obtained with approval from both the Chinese Authorities for Animal Protection and also approved by from the Ethics Committee of Dalian Medical University.

In addition, all experiments were conducted in accordance with the guidelines and regulations of Dalian Medical University. The collected Gentoo penguin carcasses were fixed and stored in a 10% formalin solution for subsequent experiments.

### Anatomical dissection in the suboccipital region

Three of the six penguin specimens were used for gross dissection. The epidermis was cut longitudinally along the dorsal midline of the specimens, and the suboccipital musculature was fully exposed. The deep suboccipital musclature, including the RCDmi, (analogous to the RCPmi muscle found in humans) and the RCDma, (analogous to the RCPma muscle also found in humans) was exposed. The deep suboccipital muscles (RCDmi and RDCma) were then both cut from the bony occipital crest to observe their connections with the dorsal atlanto-occipital membrane (DAOM, analogous to the PAOM observed in humans) and the dorsal atlanto-axial membrane. Next the DAOM was also cut along its cranial attachment to observe the connection between the DAOM and the SDM. Lateral incision of the posterior arch of the atlas was also made to observe the connection between the dorsal atlanto-axial membrane and the SDM. The photographic materials were taken with a Canon 7D camera (Canon Inc., Tokyo, Japan).

### Histological slices and staining

Three specimens were used for histological studies. Tissue samples of the occiput and the cervical region were immobilized in a 10% formalin solution for ten days. Subsequently, the specimens were transferred to Jiang Weizhong's decalcification solution for decalcification [27]. The Jiang Weizhong's decalcification solution was changed at three-day intervals until the bone was easily punctured by a needle (21 days). The decalcified tissue samples were then washed overnight in running water, and then dehydrated with increasing grades of alcohol, transparent in xylene, and then infiltrated with melted paraffin, and embedded. A rotary microtome (Leica Micro HM450; Lei Microsystems GmbH, Wetzlar, Germany) was used to cut 8-μm-thick slices. These tissue slices were then divided into three groups: Group 1 slices were used for Hematoxylin and Eosin (HE) staining, which can show the general structural features of the MDB; Group 2 slices were used for Masson trichrome staining (Masson), to detect any collagen fibers in the MDB; Group 3 slices were used for Picrosirius Red (PRS) staining, which can also detect collagen fibers in the MDB, and most importantly, collagen types can be distinguished when viewed with a polarized light microscope. Microscopic examination and photography were carried out with the Nikon ECLIPSE80i research light microscope, and multiple images of each section were stitched together using the Microsoft Image Synthesis Editor of Nikon ECLIPSE80i Image Processing and Analysis System. The results of Picrosirius Red staining were observed with a light microscope as well as a polarized light microscope.

## Results

### Anatomical dissection of the suboccipital region

The RCDmi and the RCDma muscles, located in the deep suboccipital region, were observed to be tightly connected to each other (Fig 1a). The cranial end of RCDmi muscle attached to the medial part of the occipital crest, and the caudal end attached to the dorsal side of the posterior tubercle of atlas. Moreover multiple dense fibrous tissues were observed to originate from the ventral part of the RCDmi connect to the DAOM (Fig 1b). The DAOM and the SDM were intimately connected to each other by numerous trabecular fibrous bundles (Fig 1c). The cranial end of the RCDma muscle attached to the lateral aspect of the occipital crest, and the caudal end attached to the spinous process of the axis. Some of the dense fibrous bundles originating from the ventral part of the RCDma muscle connected to the dorsal atlanto-axial membrane (Fig 2a). The dorsal atlanto-axial membrane was tightly adheret to the SDM via several dense cord-like fibrous tissue (Fig 2b). This cord-like trabecular fibrous tissue was found in all the three specimens.

### Histology studies

In the HE-stained sections, multiple dense fibers originating from the ventral anterior aspect of RCDmi pass through the atlanto-occipital interspace (Fig 3a). These fibers extend from the ventral side of the RCDmi muscle and terminate in the DAOM. The DAOM emitted dense fibrous bundles and connected with the SDM passing through several venous sinuses (Fig 3b and 3c). However, different origins of these connective fibers in the atlanto-axial interspace were found. Some of the fibrous bundles originating from the ventral part of the RCDmi connect with the dorsal atlanto-axial membrane directly (Fig 3d). Fibers originating from the ventral aspect of RCDma as well as fibers originating from the OCP fuse together then pass through the dorsal atlanto-axial membrane and terminate on the SDM (Fig 3d). These

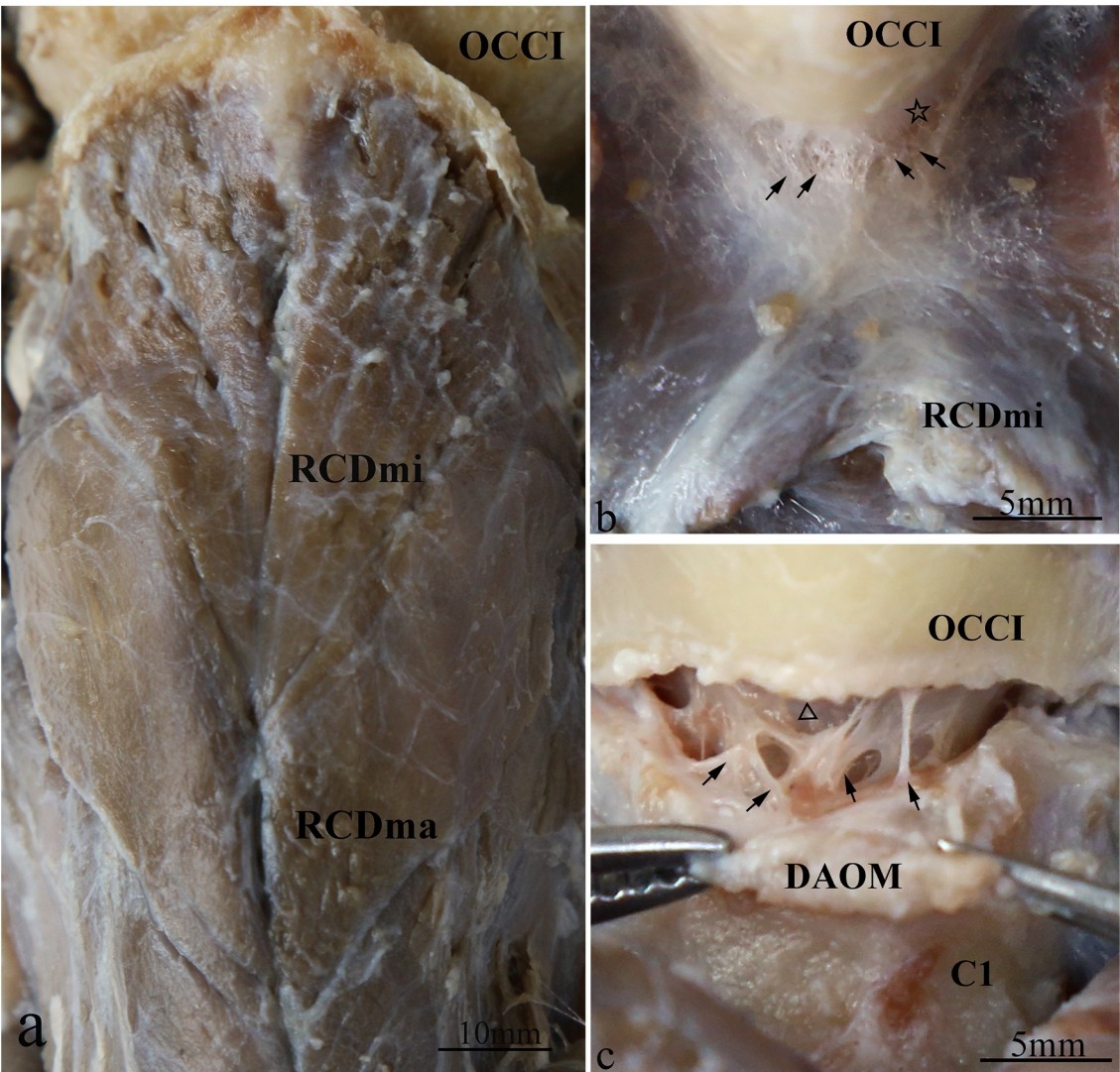

**Fig 1. Gentoo penguin (*Pygoscelis papua*), dorsal views of an anatomical dissection of the deep suboccipital space.** a: Superficial view of the deep suboccipital region. The RCDmi and the RCDma are connected tightly. b: The connection between the RCDmi and the DAOM. The ventral surface of the RCDmi is connected by multiple dense fibrous tissues (arrow) to the DAOM (hollow star). c: The connection between the DAOM and the SDM. The DAOM was separated from its cranial attachment at the foramen magnum, and the ventral surface of the DAOM is connected by trabecular fibrous bundles (arrow) to the SDM (hollow triangle). Abbreviation: OCCI = occipital bone; RCDmi = rectus capitis dorsalis minor muscle; RCDma = rectus capitis dorsalis major muscle; DAOM = dorsal atlanto-occipital membrane; SDM = spinal dura mater; C1 = posterior arch of atlas.

connective tissue fibers observed between the muscles and the SDM are analogous to the MDB fibers previously observed in humans. In the Masson-stained sections, the muscular fibers were stained red, and the MDB fibers were stained blue (Fig 4). The results of the Masson staining demonstrate that the MDB fibers present in Gentoo penguins are collagenous fibers. In the PSR-stained sections, the fibers of the MDB were stained red under when viewed the ordinary optical microscope (Fig 5a), which implies that the penguin MDB is composed primarily of collagen fibers. Viewed with the polarizing microscope, the fibrous tissues of MDB were stained either red or yellow (Fig 5b–5d). This observation implies that the MDB is composed of primarily type I collagen fibers.

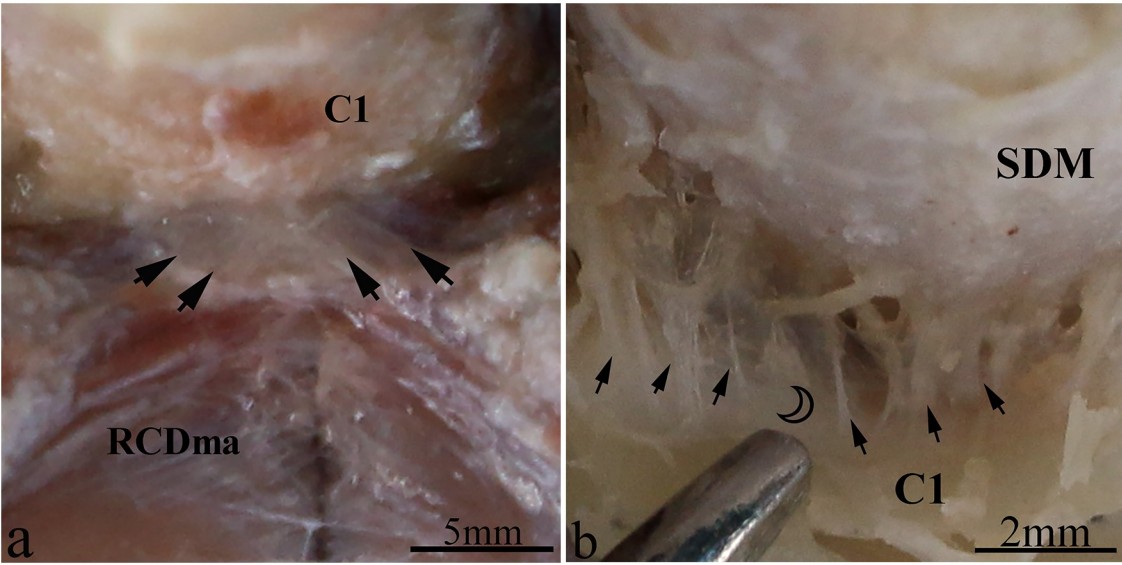

**Fig 2. Gentoo penguin, dorsal views of an anatomical dissection of the Atlanto-axial interspace.** a: The connection between the RCDma and the dorsal atlanto-axial membrane. The ventral surface of RCDma is connected by dense fibrous bundles (arrow) to the dorsal atlanto-axial membrane. b: The connection between the dorsal atlanto-axial membrane and the SDM. The dorsal atlanto-axial membrane was separated from its posterior arch of atlas, and the ventral surface of the dorsal atlanto-axial membrane is connected by dense cord-like fibrous tissues (arrow) to the SDM (hollow crescent). Abbreviation: RCDmi = rectus capitis dorsalis minor muscle; RCDma = rectus capitis dorsalis major muscle; SDM = spinal dura mater; C1 = posterior arch of atlas.

## Discussion

In 1995 Hack et al. initially proposed the concept of the MDB [1]. Numerous researchers have since studied the MDB in detail, confirming the MDB's existence in both humans [1–14] and animals [20–26]. These researchers have further hypothesized about the putative physiological functions of the MDB [8–19]. The MDB is described as a dense connective tissue bridge connecting the suboccipital musculature to the cervical SDM, while passing through the posterior atlanto-occipital and the altanto-axial interspaces. Therefore, what is unique about the penguin as a marine bird?

According to the observation of the gross anatomy of the penguin, evidenced in the present study, the deep suboccipital musculature of the Gentoo penguin consists of the RCDmi and the RCDma muscles. The cranial ends of both the RCDmi and the RCDma muscles respectively attach to the medial and lateral aspects of the occipital crest. They were intimately related to each other and not easily separated. Interestingly, it was previously found that the RCDmi muscle of the *Nephocaena phocaenoides* was located on the deep surface of the RCDma, and the RCDmi was observed originating from occiput [21]. Furthermore, the RCDmi of the sperm whale was previously found to originate from the dorsal aspect of the occipital squama [22]. This spatial distribution of the RCDmi in marine mammals (*Nephocaena phocaenoides* and sperm whale) is similar to that of the Gentoo penguin. However, the cranial attachments of both the RCDmi and RCDma muscle of the Gentoo penguin are distinct from that observed in other birds (Rock pigeons and *Gallus domesticus*) [25, 26]. The cranial aspect of the RCDmi attached below the transverse nuchal crest, and the both the RCDmi and RCDma muscle were easily detached [25, 26]. This difference may be related to the survival and feeding behavior of the penguin. These behaviors can be inferred from skull morphology [28].

In the present study, we have confirmed the existence of the MDB in the Gentoo penguin through multiple methods. In the atlanto-occipital interspace of the Gentoo penguins, the

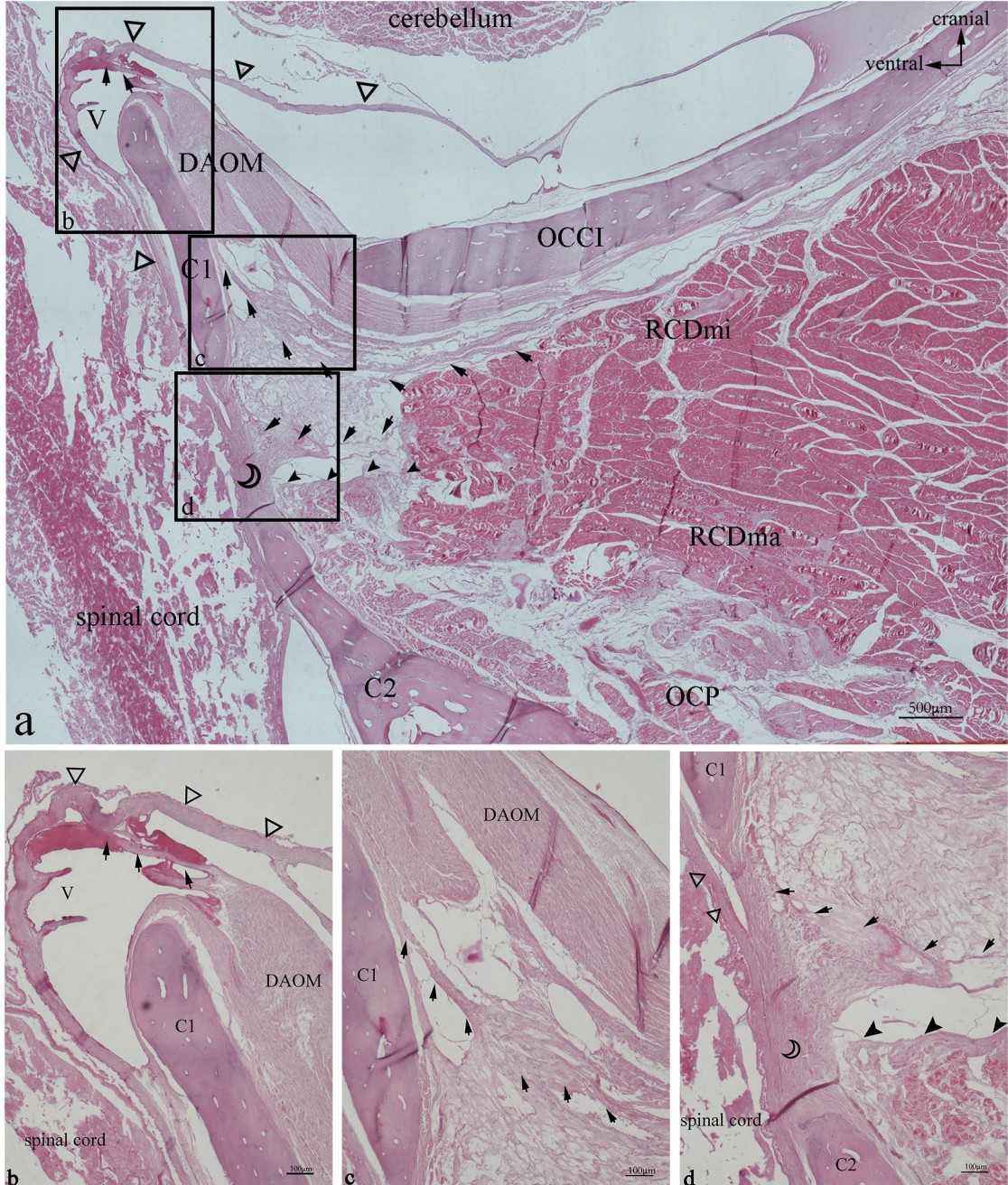

**Fig 3. Gentoo penguin, prepared sagittal section of the suboccipital region with stained with HE.** Area within square in a is shown enlarged in b, c, and d. The fibers (arrow) originate from the ventral anterior of RCDmi, extend directly into the atlanto-occipital interspace, then fuse with the DAOM, and the DAOM extends into a fiber bundle through the venous sinus to connect with the SDM (hollow triangle). In the atlanto-axial interspace, the dense fibers bundles (arrow) originating from the ventral side of the RCDmi run ventrally run ventrally, and integrate into the dorsal atlanto-axial membrane (hollow crescent) together with the dense fibers(filled arrowhead) originating from the RCDma and OCP, and send out dense fibers bundles to closely connect with the SDM. Abbreviation: OCCI = occipital bone; RCDmi = rectus capitis dorsalis minor muscle; OCP = oblique capitis posterior; DAOM = dorsal atlanto-occipital membrane; SDM = spinal dura mater; C1 = atlas; C2 = axial; V = venous sinus.

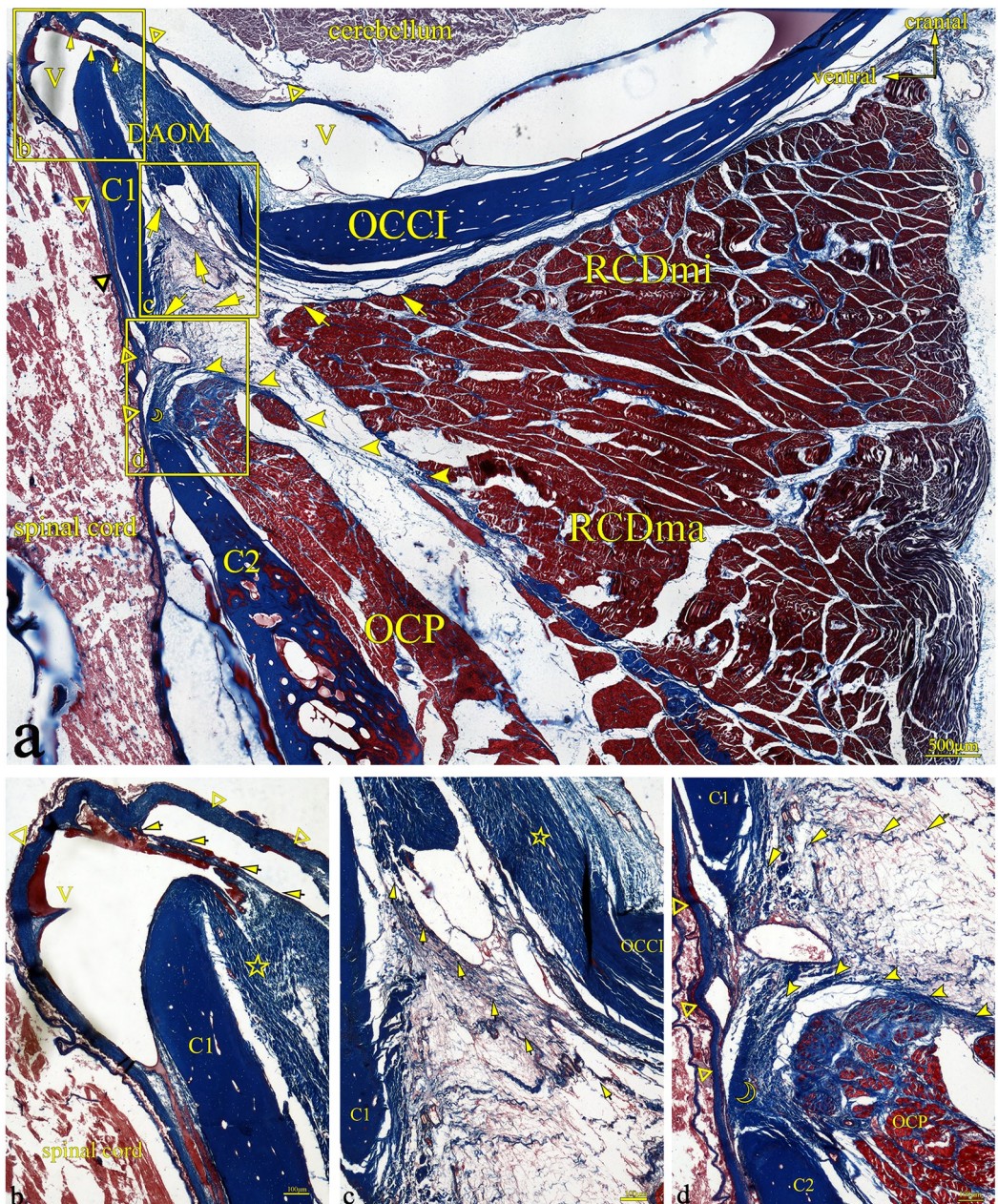

**Fig 4. Gentoo penguin, prepared sagittal section of the suboccipital region with stained with Masson.** Area within square in a is shown enlarged in b, c, and d. The fibers (arrow) originate from the ventral anterior of RCDmi, extend directly into the atlanto-occipital interspace, then fuse with the DAOM, and the DAOM extends into a fiber bundle through the venous sinus to connect with the SDM (hollow triangle). In the atlanto-axial interspace, the dense fibers bundles (arrow) originating from the ventral side of the RCDmi run ventrally, and integrate into the dorsal atlanto-axial membrane (hollow crescent) together with the dense fibers(filled arrowhead) originating from the RCDma and OCP, and send out dense fibers bundles to closely connect with the SDM. Abbreviation: OCCI = occipital bone; RCDmi = rectus capitis dorsalis minor muscle; OCP = oblique capitis posterior; DAOM = dorsal atlanto-occipital membrane; SDM = spinal dura mater; C1 = atlas; C2 = axial; V = venous sinus.

dense MDB fibers originated primarily from the ventral aspect of the RCDmi. These fibers extend from the ventral side and extend superiorly to fuse with the DAOM. Then these fibers connect with the SDM passing through the venous sinus. In the atlanto-axial interspace of the Gentoo penguin, the dense MDB fibers originate from the ventral of the RCDmi, the RCDma,

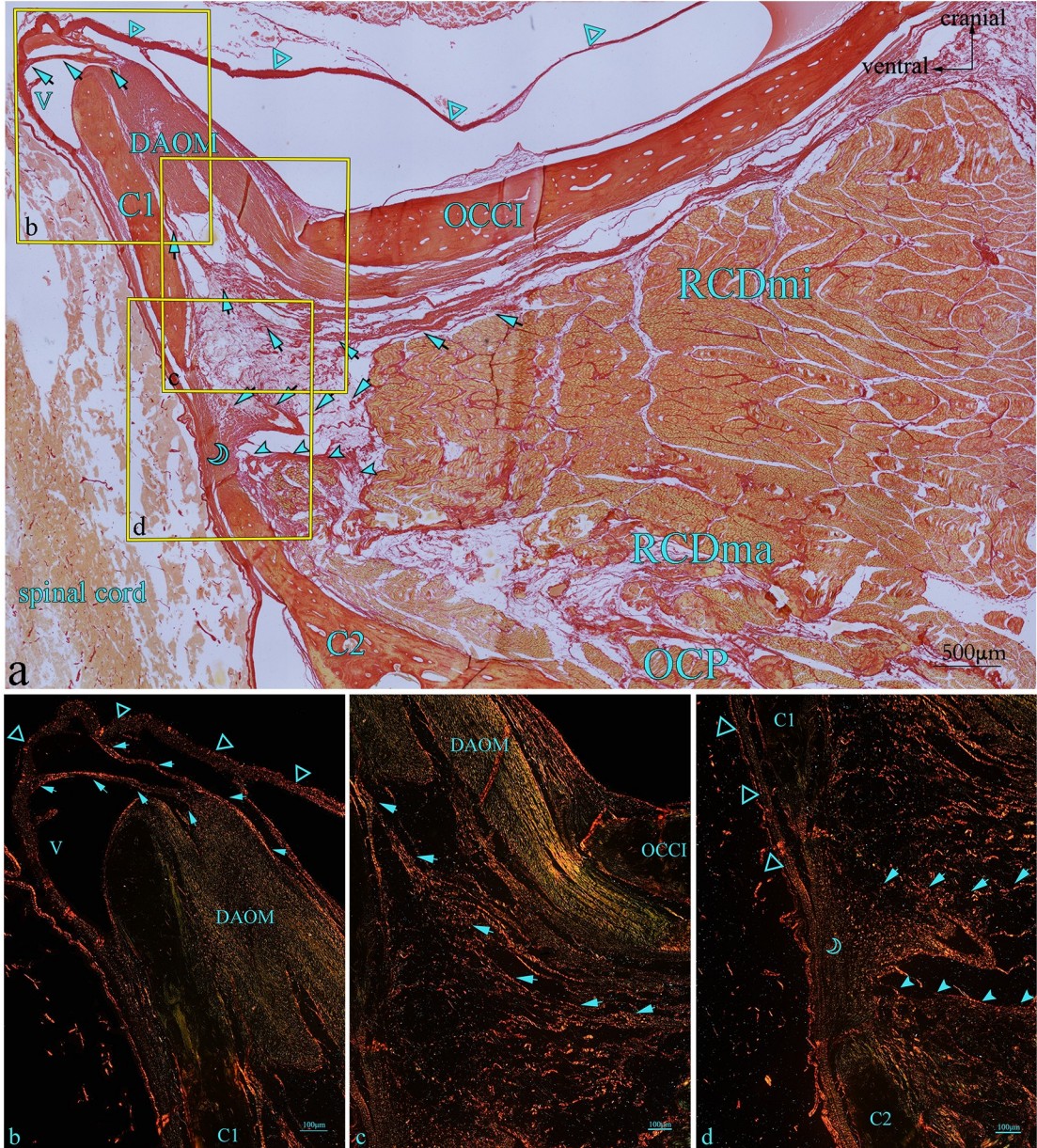

**Fig 5. Gentoo penguin, prepared sagittal section of the suboccipital region with stained with PSR-stained.** Area within square in a is shown enlarged in b, c and d. The fibers (arrow) originate from the ventral anterior of RCDmi, extend directly into the atlanto-occipital interspace, then fuse with the DAOM, and the DAOM extends into a fiber bundle through the venous sinus to connect with the SDM (hollow triangle). In the atlanto-axial interspace, the dense fibers bundles (arrow) originating from the ventral side of the RCDmi run ventrally, and integrate into the dorsal atlanto-axial membrane (hollow crescent) together with the dense fibers(filled arrowhead) originating from the RCDma and OCP, and send out dense fibers bundles to closely connect with the SDM. Abbreviation: OCCI = occipital bone; RCDmi = rectus capitis dorsalis minor muscle; OCP = oblique capitis posterior; DAOM = dorsal atlanto-occipital membrane; SDM = spinal dura mater; C1 = atlas; C2 = axial; V = venous sinus.

and the OCP, all converging to form the MDB, and then passing through the dorsal atlanto-axial membrane, which is tightly connected to the SDM. The arrangement of the fibers of the MDB, observed in the Gentoo penguin is similar to that of other birds [25, 26] and mammals [20]. Although penguins live in different conditions from that of the Rock pigeons and the

*Gallus domesticus*, they all possess an MDB. Once again, the MDB is seen as a highly conserved structure. During the course of biological evolution, structures that are not functionally important tend to degrade, and this evolutionary conservation of the MDB demonstrates the significance of MDB. In addition, the MDB observed in the Gentoo penguin presents as a dense fibrous structure that is composed of type I collagen fibers.

While, being a marine bird, penguin not only has the same type of MDB as that observed in flying birds, but also has the same spatial distribution of the suboccipital muscles like marine mammals. What then are the similarities and differences between the MDB of penguins and marine mammals? Research has confirmed that no DAOM exists in the posterior interspace between the occipital bone and atlas of *Nephocaena phocaenoides* [21]. The tendinous fibers of the RCDmi observed in the *Nephocaena phocaenoides* projected through the atlanto-occipital interspace attaching directly to the SDM. This direct connection between the RCDmi and the SDM of the *Nephocaena phocaenoides* might create a more pronounced effect of the RCDmi muscle on the SDM [21]. Furthermore, the present authors have confirmed that *sperm whales* have two different origins of their MDB: one origin is from the ODB (occipital dural bridge) which originates from the periosteal surface of the occiput and fuses with the dura mater [22], and the other bridge (MDB) originates from the RCDmi, which transmits the tension from the RCDmi to the dura mater. The MDB stabilizes the dura mater, the spinal cord, and the ODB [22]. The MDB, observed in the atlanto-occipital interspace of penguins, is primarily composed of dense connective tissue fibers originating from the RCDmi muscle. Therefore, there is a type of the MDB, originating from the RCDmi, in penguins, *sperm whales*, and *Nephocaena phocaenoides* which all pass through the atlanto-occipital interspace. Moreover, they have similar distribution of suboccipital muscles and the presence of numerous suboccipital venous sinuses. As these different animals live in similar conditions, some of their anatomic structures tend to be similar. This is referred to as the "similar effect". Therefore, as the penguin is capable of navigating in deep water, this ability maybe associated with cerebrospinal fluid circulation. This also demonstrates that the MDB is a significant and highly conserved anatomic structure.

Interestingly, the *sperm whale* has an atlas and six fused vertebra (C2-C7) [29], while their MDB exists only in the atlanto-occipital interspace, with extensive venous plexus among the MDB fibers within their atlanto-occipital interspace [22]. As *sperm whales* swim to great depths, the MDB of these animals may contribute to transferring the tensile forces generated by the suboccipital muscles to the cervical dura mater and thereby continuously alter the volume of the CSF contained within the subarachnoid space, acting as a unique mechanism to facilitate circulation of the CSF [22]. The penguin's MDB exists in both the atlanto-occipital and the atlanto-axial interspaces, and there are numerous venous sinuses among the MDB fibers. As the penguin dives to great depths, the contraction of MDB fibers via the suboccipital musculature may accelerate venous blood flow back to the heart and also promote cerebrospinal fluid circulation. According to previous research, the penguin can dive to approximately 100 meters within 3 minutes [30, 31]. Therefore, the penguin's MDB might be a key structure which helps the penguin to maintain normal cerebrospinal fluid circulation during deep dives.

In this study, the present authors validated that there are dense connective tissues (MDB) in both the atlanto-occipital and atlanto-axial interspaces of Gentoo penguin, that are composed of primarily type I collagen fibers. Therefore, this finding implies that the MDB is not only a highly conserved anatomic structure, from an evolutionary point of view, but also has its variant structure for acclimatization. These findings provide supporting evidence for research into the physiological function of the MDB.

## Supporting information

**S1 File. Staining methods.**
(DOC)

## Acknowledgments

We are grateful to Haichang Ocean Park Holdings., Ltd for provision of cadaveric specimens. Thanks to Gary D. Hack of University of Maryland School of dentistry, USA, for helping with language editing.

## Author Contributions

**Conceptualization:** Nan Zheng, Hong-Jin Sui.

**Data curation:** Cheng Chen, Nan Zheng.

**Formal analysis:** Cheng Chen, Sheng-bo Yu, Nan Zheng.

**Funding acquisition:** Nan Zheng, Hong-Jin Sui.

**Investigation:** Cheng Chen, Yan-yan Chi, Guang-yuan Tan, Bao-cheng Yan.

**Methodology:** Cheng Chen.

**Project administration:** Nan Zheng, Hong-Jin Sui.

**Resources:** Hong-Jin Sui.

**Software:** Cheng Chen.

**Supervision:** Hong-Jin Sui.

**Validation:** Hong-Jin Sui.

**Visualization:** Cheng Chen.

**Writing – original draft:** Cheng Chen, Nan Zheng.

**Writing – review & editing:** Nan Zheng, Hong-Jin Sui.

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
