## [Decision Letter · Decision Letter 0]

18 Jan 2021

PONE-D-20-39348

Existence and Features of the Myodural Bridge in Gentoo Penguins: a morphological study

PLOS ONE

Dear Dr. Chen,

Thank you for submitting your manuscript to PLOS ONE. After careful consideration, we feel that it has merit but does not fully meet PLOS ONE’s publication criteria as it currently stands. Therefore, we invite you to submit a revised version of the manuscript that addresses the points raised during the review process.

Please consider all the comments of the two reviewers during the manuscript revision.

We look forward to receiving your revised manuscript.

Kind regards,

Aldo Corriero, Ph.D.

Academic Editor

PLOS ONE

Journal Requirements:

We note that one or more of the authors are employed by a commercial company: Haichang Ocean Park Holdings.,Ltd,.

2.1. Please provide an amended Funding Statement declaring this commercial affiliation, as well as a statement regarding the Role of Funders in your study. If the funding organization did not play a role in the study design, data collection and analysis, decision to publish, or preparation of the manuscript and only provided financial support in the form of authors' salaries and/or research materials, please review your statements relating to the author contributions, and ensure you have specifically and accurately indicated the role(s) that these authors had in your study. You can update author roles in the Author Contributions section of the online submission form.

2.2. Please also provide an updated Competing Interests Statement declaring this commercial affiliation along with any other relevant declarations relating to employment, consultancy, patents, products in development, or marketed products, etc.  

Reviewers' comments:

Reviewer's Responses to Questions

**Comments to the Author**

1. Is the manuscript technically sound, and do the data support the conclusions?

Reviewer #1: Partly

Reviewer #2: Yes

2. Has the statistical analysis been performed appropriately and rigorously? 

Reviewer #1: N/A

Reviewer #2: N/A

3. Have the authors made all data underlying the findings in their manuscript fully available?

Reviewer #1: Yes

Reviewer #2: Yes

4. Is the manuscript presented in an intelligible fashion and written in standard English?

Reviewer #1: No

Reviewer #2: Yes

5. Review Comments to the Author

Reviewer #1: This is an interesting study that could provide some novel comparative interpretations of a rarely considered anatomical region in animals.

However, there are a number of aspects that need to be improved before it would be acceptable for publication. Most importantly, the discussion currently provides a very limited analysis of the specific anatomy in the target species and unique aspects might relate to specific aspects of species biology. A more thorough analysis of the comparative anatomy of the species included within the discussion and the relationship between this and the potential functional correlates is necessary.

There are numerous grammatical, typographical and formatting issues (some of which i have highlighted within the pdf document) that need to be addressed.

The resolution of the photographs within the figures is currently not adequate, and many of the labels are not legible.

Reviewer #2: Thank you for this addition to the literature regarding the myodural bridges.

In the abstract, cut the sentence "The present authors suggest..." as it does not add to the main effort of this submission. Also, move the sentence "While foraging..." to follow the statement about the Gentoo penguins being the fastest diving birds.

In the introduction, there are several myodural bridges described, so add the word "Each" at the beginning of the second sentence to make it, "Each myodural bridge (MDB) is..."

In the methods on "histological slices and staining" you use both "Picrosirius Red (PRS)" and "Picric acid-Sirius red" to describe the stain. While the two terms are synonymous, you might want to pick one version and use it consistently in this paper.

In the results, the legends for Figures 3, 4, and 5 all have the same typo. I would suggest "In the atlanto-axial interspace, the dense fiber bundles (arrow) originating from the ventral SIDE OF THE RCDmi run ventrally" or something similar. I use the capital letters only for emphasis, they should be lower case in the revision.

In the Discussion, there should be a space between "In" and "1995" in the first sentence. You might want to define "natatores" as swimming birds since some readers (myself included) may not already be familiar with the term. Also, was that sentence meant to end with a question mark? That would make more sense in context. In the fourth paragraph, third sentence, change "found" to "exists". In the fifth paragraph, fourth sentence, change "ocean" to "depths".

6. PLOS authors have the option to publish the peer review history of their article (what does this mean?). If published, this will include your full peer review and any attached files.

Reviewer #1: No

Reviewer #2: **Yes: **Peter J. Ward

---

## [Author Response · Author response to Decision Letter 0]

16 Mar 2021

Reviewer #1: 

1. However, there are a number of aspects that need to be improved before it would be acceptable for publication. Most importantly, the discussion currently provides a very limited analysis of the specific anatomy in the target species and unique aspects might relate to specific aspects of species biology. A more thorough analysis of the comparative anatomy of the species included within the discussion and the relationship between this and the potential functional correlates is necessary.

Thanks for your advice, your advice of great value for our future study. At present, we only carry out a morphological study about the Gentoo penguin of "myodural bridge" to prove that "myodural bridge" universal existence and might have an impotant finction. For the next stage, related functional experiments will be carried out on the morphological abnormalities of different birds. Because "myodural bridge" in different kinds of vertebrates may have diffent morphologial characters due to diferent life style and living envionment. This will be an impotant study direction in our team.

2. There are numerous grammatical, typographical and formatting issues (some of which i have highlighted within the pdf document) that need to be addressed.

 minor remarks:

Page 3

1.rephrase for clarity/style; e.g., The myodural bridge (MDB) is a...

Line 40: The myodural bridge (MDB) is an anatomical structure connecting a suboccipital muscle (RCPmi) to the cervical spinal dura mater (SDM) in humans was identified in the atlanto-occipital interspace by Hack et al. (1995) [1].

2.year required

Line 42: ....by Hack et al. (1995) [1].

3.delete,....With these in-depth studies, many researchers now speculate that the MDB may play a significant role in physiological functions.

Line 46: ....and the nuchae ligament (NL) [2-7]. With these in-depth studies, many researchers now speculate that the MDB may play a significant role in physiological functions. It has been proposed....

4.delte....s ....

Line 50: ....play an important role in modulating ....

5. change "rencent" to "also"

Line 51: According to also reports,

Page 4

1. italics not required here

Line 55: ....and sperm whales....

Line 56: ....Siamese crocodile and....

Line 57: ....Rock pigeons and....

2. insert binomial name

Line 64: Gentoo penguins (Pygoscelis papua) 

3. fixed/preserved?

Line 72: ...."immobilized" was changed to "fixed and stored"

Page 5

1. insert relevant reference literature

Line 90: The Jiang Weizhong’s [31] 

One literature has been added in this paper

[31]. Liu ZD, Fan QY, Qiu XX, Jiang WZ. 1998 . Morphological observation of bone remodeling in adult dogs. －Journal of the Fourth Military Medical University. (01), 108-109. doi:CNKI:SUN:DSJY.0.1998-01-045.

Page 7

1.was tightly adherent... (past-tense - inconsistent use of tense throughout)

Line 116: ....The dorsal atlanto-axial membrane was tightly adheret to....

2.insert binomial name in figure captions

Line 140: Gentoo penguins (Pygoscelis papua) 

3. "membrane" and "was"

Line 132: membrane was

Page 11

1. Rephrase for clarity

Line 194: Therefore, what is unique about the penguin as a marine bird？

2. formatting issues

Line 200: the Nephocaena phocaenoides 

Line 202: sperm whale 

Line 204: and sperm whale

Line 206: Rock pigeons and

3. as described in previous research? references required here

Line 207: other birds (Rock pigeons and Gallus domesticus) [23, 24]. 

[23]. Chukwuemeka Samuel Okoye, Zheng N, Yu SB, Sui HJ. 2018. The myodural bridge in the common rock pigeon ( Columbia livia) : Morphology and possible physiological implications.－J Morphol. 279(10):1524-1531. doi: 10.1002/jmor.20890.

[24]. Dou YR, Zheng N, Gong J, Tang W, Chukwuemeka Samuel Okoye, Zhang Y, Chen YX , Zhang Y, Pi SY, Qu LC, Yu SB, Sui HJ. 2019. Existence and features of the myodural bridge in Gallus domesticus: indication of its important physiological function.－Anat Sci Int. 94(2):184-191. doi: 10.1007/s12565-018-00470-2.

4. Line 209: in what ways? more specific hypotheses based on some documented relevant biological aspects of penguins would be useful here.

At present, we distinguish differences, mainly on the basis of gross anatomy. For the next stage, related functional experiments will be carried out on the morphological abnormalities of different birds.

Page 12

1. formatting issues

Line 220: Rock pigeons

Page 17

1. formatting issues

Line 330: Nephocaena phocaenoides

Page 18

1. formatting issues

Line 340: Trachemys scripta elegans

2.formatting issues

Line 343: Columbia livia

3. formatting issues

Line 347: Gallus domesticus

4. formatting issues

Line 358: Pygoscelis 

Page 19

1. formatting issues

Line 362: Pygoscelis papua

2.formatting issues

Line 364: Pygoscelis papua

3. The resolution of the photographs within the figures is currently not adequate, and many of the labels are not legible.

Thank you for your advice. The resolution and label of the image have been modified.

I have uploaded the image file to the "Pre-flight Analysis and Transformation Engine (PACE) Digital Diagnostic Tool" for inspection. I have deepened the color of the image.

Reviewer #2: 

1. In the abstract, cut the sentence "The present authors suggest..." as it does not add to the main effort of this submission. Also, move the sentence "While foraging..." to follow the statement about the Gentoo penguins being the fastest diving birds.

Page 2

Line 19: ....interspaces. The present authors suggest that the MDB has important physiological functions in humans. The existence of the MDB....

Line 22: ....660 feet. While foraging, this penguin is able to reach speeds of up to 22 miles per hour. Gentoo penguins are also....

2. In the introduction, there are several myodural bridges described, so add the word "Each" at the beginning of the second sentence to make it, "Each myodural bridge (MDB) is..."

Page 3

Line 42: ....Each myodural bridge (MDB) is now described as a fibrous....

3. In the methods on "histological slices and staining" you use both "Picrosirius Red (PRS)" and "Picric acid-Sirius red" to describe the stain. While the two terms are synonymous, you might want to pick one version and use it consistently in this paper.

Page 6

Line 103: "Picric acid-Sirius red " was changed to "Picrosirius Red (PRS)" here and use it consistently in this paper.

....The results of Picrosirius Red staining were observed with a light microscope as well as a polarized light microscope.

4.In the results, the legends for Figures 3, 4, and 5 all have the same typo. I would suggest "In the atlanto-axial interspace, the dense fiber bundles (arrow) originating from the ventral SIDE OF THE RCDmi run ventrally" or something similar. I use the capital letters only for emphasis, they should be lower case in the revision.

Page 9

Line 159: ....the ventral side of the RCDmi run ventrally run ventrally....

Line 170: ....the ventral side of the RCDmi run ventrally run ventrally....

Line 181: ....the ventral side of the RCDmi run ventrally run ventrally....

5.In the Discussion, there should be a space between "In" and "1995" in the first sentence. You might want to define "natatores" as swimming birds since some readers (myself included) may not already be familiar with the term. Also, was that sentence meant to end with a question mark? That would make more sense in context. In the fourth paragraph, third sentence, change "found" to "exists". In the fifth paragraph, fourth sentence, change "ocean" to "depths".

Page 10

Line 188: In 1995 Hack et al. ....

Page 11

Line 194: "natatores" was changed to "main bird" .

....Therefore, what is unique about the penguin as a marine bird？

Page 12

Line 228: "found" was changed to "exists"

Page 13

Line 255: "ocean" was changed to "depths".

Academic editor

Manuscripts have been ensured to meet PLoS ONE stylistic requirement, including document naming.

Funding Statement

Funding: This research was supported by the National Natural Science Foundation of China (NSFC31871213 to HJS); and Department of Education of Liaoning Province (LZ2020048 to NZ). The funders had no role in study design, data collection and analysis, decision to publish, or preparation of the manuscript.

Competing Interests Statement 

Haichang Ocean Park Holding Co., Ltd. is responsible for the breeding of Gentoo Penguin (Pygoscelis Papua) and provides the basic data related to the penguin specimens for this research group. Another part of their work is the study and popularization of science related to penguins. The authors have declared that no competing interests exist.

This does not alter our adherence to PLOS ONE policies on sharing data and materials.

---

## [Decision Letter · Decision Letter 1]

24 Mar 2021

Existence and Features of the Myodural Bridge in Gentoo Penguins: a morphological study

PONE-D-20-39348R1

Dear Dr. Chen,

We’re pleased to inform you that your manuscript has been judged scientifically suitable for publication and will be formally accepted for publication once it meets all outstanding technical requirements.

Kind regards,

Aldo Corriero, Ph.D.

Academic Editor

PLOS ONE

Additional Editor Comments (optional):

Reviewers' comments:

Reviewer's Responses to Questions

**Comments to the Author**

1. If the authors have adequately addressed your comments raised in a previous round of review and you feel that this manuscript is now acceptable for publication, you may indicate that here to bypass the “Comments to the Author” section, enter your conflict of interest statement in the “Confidential to Editor” section, and submit your "Accept" recommendation.

Reviewer #2: All comments have been addressed

2. Is the manuscript technically sound, and do the data support the conclusions?

Reviewer #2: Yes

3. Has the statistical analysis been performed appropriately and rigorously? 

Reviewer #2: N/A

4. Have the authors made all data underlying the findings in their manuscript fully available?

Reviewer #2: Yes

5. Is the manuscript presented in an intelligible fashion and written in standard English?

Reviewer #2: Yes

6. Review Comments to the Author

Reviewer #2: All my comments from the first round of review were incorporated appropriately. I have no additional concerns.

7. PLOS authors have the option to publish the peer review history of their article (what does this mean?). If published, this will include your full peer review and any attached files.

Reviewer #2: **Yes: **Peter J Ward

---

## [Editor Report · Acceptance letter]

29 Mar 2021

PONE-D-20-39348R1 

Existence and Features of the Myodural Bridge in Gentoo Penguins: a morphological study 

Dear Dr. Chen:

I'm pleased to inform you that your manuscript has been deemed suitable for publication in PLOS ONE. Congratulations! Your manuscript is now with our production department. 

Kind regards, 

on behalf of

Dr. Aldo Corriero 

Academic Editor

PLOS ONE